# Noncoding RNAs in Hepatocellular Carcinoma: Potential Applications in Combined Therapeutic Strategies and Promising Candidates of Treatment Response

**DOI:** 10.3390/cancers16040766

**Published:** 2024-02-13

**Authors:** Clara Vianello, Elisa Monti, Ilaria Leoni, Giuseppe Galvani, Catia Giovannini, Fabio Piscaglia, Claudio Stefanelli, Laura Gramantieri, Francesca Fornari

**Affiliations:** 1Centre for Applied Biomedical Research—CRBA, University of Bologna, 40138 Bologna, Italy; clara.vianello2@unibo.it (C.V.); elisa.monti10@unibo.it (E.M.); ilaria.leoni5@unibo.it (I.L.); giuseppe.galvani2@unibo.it (G.G.); 2Department for Life Quality Studies, University of Bologna, 47921 Rimini, Italy; claudio.stefanelli@unibo.it; 3Department of Medical and Surgical Sciences, University of Bologna, 40128 Bologna, Italy; catia.giovannini4@unibo.it (C.G.); fabio.piscaglia@unibo.it (F.P.); 4Division of Internal Medicine, Hepatobiliary and Immunoallergic Diseases, IRCCS Azienda Ospedaliero-Universitaria di Bologna, 40138 Bologna, Italy; laura.gramantieri@aosp.bo.it

**Keywords:** HCC, noncoding RNA, microRNA, TKI, ICI, biomarkers

## Abstract

**Simple Summary:**

Hepatocellular carcinoma (HCC) is a leading cause of cancer-related morbidity and mortality. In the last decade, a breakthrough in the treatment landscape of HCC has been experienced. The unprecedented number of therapeutic options for advanced stages has made the selection of sequence strategies more complex and the need for biomarkers of treatment response or tumor escape more urgent. The understanding of molecular events leading to drug resistance has identified noncoding RNAs as promising therapeutic targets. Preclinical studies testing the combined efficacy of noncoding RNAs and clinically available drugs represent a crucial step to prevent/limit the onset of drug resistance in advanced cases.

**Abstract:**

The incidence of hepatocellular carcinoma (HCC) is increasing, and 40% of patients are diagnosed at advanced stages. Over the past 5 years, the number of clinically available treatments has dramatically increased for HCC, making patient management particularly complex. Immune checkpoint inhibitors (ICIs) have improved the overall survival of patients, showing a durable treatment benefit over time and a different response pattern with respect to tyrosine kinase inhibitors (TKIs). Although there is improved survival in responder cases, a sizeable group of patients are primary progressors or are ineligible for immunotherapy. Indeed, patients with nonviral etiologies, such as nonalcoholic steatohepatitis (NASH), and alterations in specific driver genes might be less responsive to immunotherapy. Therefore, improving the comprehension of mechanisms of drug resistance and identifying biomarkers that are informative of the best treatment approach are required actions to improve patient survival. Abundant evidence indicates that noncoding RNAs (ncRNAs) are pivotal players in cancer. Molecular mechanisms through which ncRNAs exert their effects in cancer progression and drug resistance have been widely investigated. Nevertheless, there are no studies summarizing the synergistic effect between ncRNA-based strategies and TKIs or ICIs in the preclinical setting. This review aims to provide up-to-date information regarding the possible use of ncRNAs as therapeutic targets in association with molecular-targeted agents and immunotherapies and as predictive tools for the selection of optimized treatment options in advanced HCCs.

## 1. Introduction

The incidence of hepatocellular carcinoma (HCC) is increasing globally, representing the sixth most common cancer and the third leading cause of cancer-related death in 2020 [1]. Over the past decade, there has been a remarkable breakthrough in the availability of systemic treatment options for advanced HCC. Despite the robust efforts in clinical trials testing several molecular-targeted compounds, after its approval in 2007 [2], sorafenib remained the only systemic treatment for patients at advanced stages until 2017, when regorafenib was granted approval in the second-line setting [3]. Subsequently, other positive phase III studies for tyrosine kinase inhibitors (TKIs) led to the approval of lenvatinib in the first line [4] and cabozantinib and ramucirumab in the second line after sorafenib progression [5,6]. Remarkably, the advent of immune checkpoint inhibitors (ICIs) in the oncologic field further revolutionized the management of HCC patients, with the IMbrave150 phase III trial demonstrating the superior efficacy of the atezolizumab–bevacizumab combination in terms of overall survival (OS) with respect to sorafenib, becoming the new front-line standard of care for HCC. More recently, a second immunotherapy-based combination, durvalumab plus tremelimumab, was qualified as a further front-line regimen [7]. Since the advent of immunotherapy, overall survival has gradually increased over time, showing benefits for patients with sensitive tumors and preserved liver function, the latter being a clinical requirement for treatment eligibility. On the contrary, vascular disorders and arterial hypertension may prevent the use of ICIs. Sorafenib and lenvatinib represent the first-line treatments of choice for patients not eligible for immunotherapy [8]. As no head-to-head comparisons are available for all the first-line treatments, the recommendation for the most appropriate choice and sequence relies only on the analysis of clinical, radiological, and biochemical profiles of the patient. In this scenario, the identification of biomarkers for patient stratification, for the delivery of sequential treatment lines, and for the optimization of clinical outcomes remains an urgent matter to be addressed in HCC research. Indeed, except for ramucirumab, for which elevated alfa-fetoprotein (AFP) levels (>400 ng/mL) are used to select patients, no biomarkers identify responder cases.

Despite substantial improvements in survival outcomes associated with sorafenib, most advanced patients do not derive a durable benefit from immunotherapy regimens. Remarkably, the etiology of HCC seems to affect the immune system response, impairing the efficacy of immunotherapy. In particular, nonalcoholic steatohepatitis (NASH)-derived HCCs were shown to be less responsive to ICIs, probably due to the aberrant activation of CD8+ T cells causing tissue damage and impaired immune surveillance [9]. Interestingly, driver mutations hindered the response to both TKIs and ICIs. PI3K–mTOR pathway alterations and the aberrant activation of WNT/β-catenin signaling were associated with shorter OS in patients treated with sorafenib and immunotherapy, respectively [10]. Although several studies gave a better understanding of molecular mechanisms involved in the onset of drug resistance in HCC with a particular focus on noncoding RNAs (ncRNAs), no drugs targeting tumor-associated ncRNAs have entered the clinical practice so far [11,12].

Here, we will describe the most recent preclinical studies, employing at least one animal model, testing the synergistic effect between ncRNAs (microRNAs, long noncoding RNAs, and circular RNAs) and TKIs or ICIs, and providing the rationale for unconventional combination strategies. As the lack of predictive biomarkers for successful patient stratification remains an open issue, the last part of this review will focus on ncRNAs, and especially microRNAs, as possible circulating candidates of treatment response and tumor escape in HCC.

## 2. Role of Noncoding RNAs in Hepatocarcinogenesis

Although the most studied sequences are those of protein-coding genes, they account for only 1–2% of the human genome. Indeed, the vast majority of the human genome encodes for ncRNAs [13]. Noncoding sequences can be transcribed into structural RNAs (rRNA, tRNA) and regulatory RNAs with essential roles in the fine-tuning of gene expression and genome organization, comprising small, medium, and long noncoding RNAs. Since their discovery, the central role of these “supposedly inert sequences” has become increasingly evident; they are transcribed into a plethora of different molecules, in both physiologic and pathologic conditions, resulting in their involvement in in most human diseases [14]. The deregulation of ncRNAs is associated with genome structural modifications or copy number variations, as well as with epigenetic or transcription factor alterations. The study by Calin et al. reported the localization of most of the aberrantly expressed microRNAs (miRNAs) within cancer-associated fragile sites of the genome. In the last two decades, the myriad of studies on miRNA activities in human diseases clearly proved their deregulation as a common cancer hallmark [15]. Recently, long noncoding RNAs (lncRNAs) and circular RNAs (circRNAs) are gaining attention in cancer research, too. Key studies on the biologic effects of miRNAs, lncRNAs, and circRNAs in hepatocarcinogenesis are reported below.

Long noncoding RNAs are >200 nucleotide-long molecules, and they regulate several cellular processes such as proliferation, differentiation, and development. They can positively or negatively regulate gene expression by acting as signals, decoys, guides, or scaffolds. They are also referred to as competing endogenous RNAs (ceRNAs), modulating protein translation and other ncRNAs, acting as miRNA sponges, or regulating small nucleolar RNAs. Recent studies have also identified lncRNAs bearing open reading frames; thus, they are able to codify for proteins [16]. In this regard, the liver-specific lncRNA AC115619 is downregulated in HCC and represents an independent factor of a poor outcome. It encodes a micropeptide, named AC115619-22aa, that plays a crucial role in the regulation of tumor progression by blocking the formation of the m6A methylation complex, which regulates the expression of key tumor suppressor (TS) genes. Strikingly, a formulation containing synthetic AC115619-22aa coupled with injectable polymeric hydrogels exerts antitumor effects in xenograft and patient-derived xenograft (PDX) mice, as well as in patient-derived organoid (PDO) models, showing its potential to become a therapeutic strategy for HCC [17]. Qualitative or quantitative alterations of lncRNAs have been described in HCC, contributing to cancerous phenotypes (e.g., metabolic reprogramming, persistent proliferation, metastasis, migration, accelerated angiogenesis, epithelial-to-mesenchymal transition, and apoptotic cell death evasion). Our group previously identified a signature of lncRNAs differentially expressed in HCC specimens versus matched cirrhotic livers [18]. Two lncRNAs (CASC9 and LUCAT1) were shown to be upregulated and one (LINC01093) downregulated in tumor tissues. Notably, a trend towards a decrease in CASC9 and LUCAT1 was observed from healthy liver to cirrhosis without HCC and to cirrhosis complicated by HCC, in line with its possible contribution to hepatocarcinogenesis. In addition, higher CASC9 and LUCAT1 levels are associated with tumor recurrence and the more aggressive properties of HCC cells. A deeper understanding of the molecular mechanism underlying the deregulation of lncRNAs would provide new insights into cancer progression, and this is crucial to the discovery of new therapeutic agents in HCC [19]. To give a couple of examples, the lncRNA CEBPA-DT is upregulated in HCC with distant metastasis and is associated with poor prognosis. Cai et al. demonstrated that CEBPA-DT induced the epithelial–mesenchymal transition (EMT) through the upregulation of Snail1 by promoting the nuclear translocation of β-catenin [20]. Similarly, the lncRNA FTO-IT1 is overexpressed in HCC, and it emerged as a glycolysis-associated lncRNA influencing the metabolic shift of cancer cells by increasing their glycolytic capacity and proliferation rate. Its effect is mediated by FTO stabilization, which in turn increases the expression of GLUT1 (glucose transporter 1) and PKM2 (pyruvate kinase M1/2) glycolytic genes in HCC cells. FTO-IT1 silencing in vivo with lentiviral vectors gave rise to smaller tumor masses and reduced FTO, GLUT1, and PKM2 expression [21]. Interestingly, HCC-specific lncRNAs regulate the activity of crucial driver genes, such as TP53 and CTTNB1 (Catenin Beta 1), which are among the most mutated genes in this disease. An example is represented by the lncRNA PSTAR, which is induced upon genotoxic and nongenotoxic stimuli, is downregulated in the tumor tissue, and acts as a TS gene by transactivating p53 signaling [22]. Moreover, PSTAR behaves as a broad-spectrum deregulated gene, which deserves attention in various cancer types. Further studies investigating mechanisms up- and downstream of PSTAR deregulation may yield new treatment strategies for HCC, especially for patients with a p53-intact pathway. In fact, due to HCC heterogeneity, it is necessary to remember the importance of stratifying patients according to their genetic background with a view to heading toward more effective and personalized therapies. Yuan and coworkers identified the upregulation of the lncRNA DANCR in HCC when compared with corresponding adjacent livers in two Asian cohorts [23]. Kaplan–Meier analysis revealed an association between higher DANCR levels and more frequent recurrence and poorer survival. Cox’s proportional hazards regression analysis indicated high DANCR expression as an independent predictor for HCC prognosis. From a molecular point of view, DANCR association with CTNNB1 3’-UTR blocked the repressing effect of three TS miRNAs (miR-214, miR-320a, miR-199a) on CTNNB1 mRNA. Xenograft and liver orthotopic mouse models demonstrated the antitumor efficacy of adeno-associated virus vector strategies for DANCR silencing, suggesting this lncRNA as a potential prognostic marker and a therapeutic target for HCC. Because DANCR stabilizes β-catenin mRNA, its action might be independent from Wnt-signaling activation and mutation in exon3 of CTNNB1, suggesting that therapeutic options exploiting its silencing might be suitable for both β-catenin mutated and WT cases. These studies suggested that cancer-associated lncRNAs may be used as potential therapeutic targets and predictive biomarkers in HCC.

Circular RNAs represent another class of ncRNAs generated by the back-splicing of linear transcripts, resulting in a circular structure that confers resistance to exonuclease activity. CircRNAs activate or repress gene expression, act as miRNA or protein sponges, enhance protein activity by forming circRNA–protein complexes acting as a scaffold, or sequester proteins in specific cellular compartments [24]. CircRNAs can also encode for polypeptides, as in the case of the liver-specific circZKSCAN1 encoding the secretory peptide circZKSaa, which sensitizes HCC cells to sorafenib by interfering with the mTOR axis, promoting its ubiquitination and degradation. CircZKSCAN1 expression is lower in tissue and serum specimens of HCC patients, suggesting that it could potentially serve as a diagnostic biomarker [25]. Another circRNA modulating the AKT/mTOR (mammalian target of rapamycin) axis is circMDK, which is upregulated in HCC and correlates with poor survival [26]. Mechanistically, circMDK sponges miR-346 and miR-874-3p to upregulate ATG16L1 (Autophagy-Related 16 Like 1), activating the AKT/mTOR signaling pathway to promote cell proliferation, migration, and in vivo tumorigenesis. A formulation containing poly β-amino esters was synthesized to favor the delivery of circMDK siRNA in four liver tumor models (subcutaneous, metastatic, orthotopic, and PDX), where it showed specific antitumor effects, offering a nanotherapeutic approach for the treatment of HCC. has_circRNA_104348 is also upregulated in liver tumors, particularly at advanced stages, and correlates with poor prognosis. Mechanistically, hsa_circRNA_104348 exerts its biological function by sponging miR-187-3p, which in turn regulates the Rho effector RTKN2 (Rhotekin 2), promoting proliferation, migration, and invasion through Wnt/β-catenin signaling pathway activation. As a proof of concept, knockdown of hsa_circRNA_104348 inhibits liver tumorigenesis and lung metastasis in xenograft mice [27]. Notably, circRPN2 is downregulated in HCC patients with postoperative metastasis or recurrence [28]. Multivariate Cox regression analyses showed circRPN2 as an independent predictor for OS and recurrence-free survival in HCC. Regarding the underlying mechanisms, downregulation of circRPN2 promotes ENO1 (enolase 1) activation, triggering the glycolytic shift of HCC cells through the AKT/mTOR pathway. Additionally, circRPN2 acts as a ceRNA for miR-183-5p, increasing FOXO1 (Forkhead Box O1) expression, which blocks tumor progression and glucose metabolism. The clinical significance of this outstanding study is that circRPN2 is a potential prognostic biomarker and therapeutic target in HCC. CircRHOT1 is upregulated in HCC and is associated with decreased OS and disease-free survival (DFS). Specifically, circRHOT1 promotes HCC cell growth, invasion, and tumor formation via recruiting the chromatin remodeling factor TIP60 to the NR2F6 (Nuclear Receptor Subfamily 2 Group F Member 6) promoter, thus triggering its transcription. In turn, NR2F6 modulates gene expression by recognizing DNA response elements linked to the regulation of adaptive immunity [29]. Another study identified the upregulation of circIPO11 in tumor tissues and cancer stem cells, where it activates the Hedgehog pathway by recruiting TOP1 (topoisomerase 1) to the promoter region of the transcription factor GLI1 (GLI family zinc finger 1). A knockout mouse model showed that circIPO11 depletion is able to suppress tumor development following carcinogen administration [30]. Recently, the exosome-mediated transfer of circRNAs is emerging as a novel mechanism in cancer progression. As an example, exosomal circRNA-100338 exerts a pro-invasive role in HCC by increasing angiogenesis, as demonstrated in preclinical models where exosomal circRNA-100338 promotes HUVEC cell proliferation, permeability, and tube formation while enhancing angiogenesis and tumor metastasis in vivo. High circRNA-100338 serum levels post-hepatectomy may predict pulmonary metastasis and poor survival in HCC [31], confirming the role of circRNAs in cancer development and aggressiveness. In summary, these studies suggest circRNAs as possible diagnostic and prognostic markers in HCC and reveal that circRNA targeting is an effective anticancer strategy.

MicroRNAs are short noncoding RNAs that are 22–25 nucleotides long and were first identified in C. elegans 30 years ago. They play key roles in the main biological processes such as cell proliferation, differentiation, and embryonic development, and they also have tissue-specific functions. As concerns miRNAs’ mechanism of action, they bind to the 3’-UTR regions of their target mRNAs, repressing their translation or triggering their degradation [32]. They can also act as intercellular communication molecules when secreted into extracellular vesicles [33]. For instance, HCC-derived exosomal miR-21 contributes to tumor progression by converting hepatic stellate cells to cancer-associated fibroblasts through PTEN downregulation, favoring tumor progression by enhancing neoangiogenesis [34]. In tumors, miRNAs can act as (TS) or oncogenes based on tumor type, stadium, and tumor microenvironment [35]. Moreover, depending on the basal expression levels of core targets or mutational background, they can act as either TSs or oncogenes not only in different tumor types but also within the same tumor, as we previously reported for miR-221 and miR-30e-3p in HCC [36,37]. MiRNAs are dysregulated not only in tumor samples but also in the surrounding liver, where chronic liver diseases (e.g., cirrhosis) may contribute to their precancerous deregulation [38].

MiRNA signatures of human tumors are associated with the diagnosis, staging, progression, prognosis, and response to treatment [39]. Others and our group firstly reported genomewide microarray profiling, identifying HCC-specific miRNAs associated with risk factors, metastasis, and oncogene/TS alterations [40,41,42]. Remarkably, the hepato-specific miR-122 was demonstrated to promote HCV replication [43] and represented the first miRNA to be silenced in vivo by chemically modified oligonucleotides in rodents and nonhuman primates [44,45]. MiRNAs can also modulate the metabolic reprogramming of HCC cells, as in the case of miR-342-3p, whose expression is high in regressing tumors. Mechanistically, miR-342-3p targets the lactate transporter MCT1 in HCC cells, thus affecting their lactate intake. In vivo, miR-342-3p delivery improved animal survival, highlighting its promising therapeutic potential [46]. Other studies confirmed miRNA modulation in vivo as an effective strategy to slow down HCC progression or prevent tumor development [47,48], opening the path towards the use of miRNAs as promising therapeutic candidates. Given their potential clinical applications, miRNAs have been the focus of cancer research in the last 20 years. Despite the early termination of the first miRNA-based clinical trial in the oncologic field [49], currently, several clinical trials using miRNAs as diagnostic biomarkers or therapeutic targets in cancer are underway [50]. 

In summary, a deeper understanding of ncRNAs as actionable targets in the preclinical setting will allow us to pinpoint mechanisms of drug resistance and synergies among treatments and to identify biomarkers for patient stratification and therapeutic sequences in HCC.

## 3. Combination of Noncoding RNA-Based Strategies with TKIs in HCC

Sorafenib and lenvatinib represent the two first-line treatments for patients not eligible for immunotherapy [51]. Due to sorafenib being the only systemic drug for advanced patients for almost a decade [2], a considerable number of the literature articles rely on molecular mechanisms underlying deregulated ncRNAs, contributing to sorafenib resistance. On the other hand, lenvatinib’s entry into clinical practice just preceded the accelerated approval of “Atezo/Beva” combination therapy for the treatment of advanced HCC, directing the interest of the scientific community toward immunotherapy, with few studies analyzing the effect of ncRNAs following lenvatinib administration. In the next two chapters, we will address the role of ncRNAs in the onset of TKI resistance, focusing on preclinical studies reporting the evaluation of combined strategies in at least one animal model.

### 3.1. Noncoding RNAs and Sorafenib Combination Improves the Therapeutic Response

Noncoding RNAs are often deregulated in HCC and are extensively involved in the modulation of molecular mechanisms leading to sorafenib resistance, such as hypoxia, autophagy, metabolic reprogramming, and activation of oncogenic pathways [12]. Sorafenib is an oral TKI that blocks tumor cell proliferation by targeting Raf/MEK/ERK signaling at the level of Raf kinase and exerts an antiangiogenic effect by targeting vascular endothelial growth factor receptor-2/-3 (VEGFR-2/-3), and platelet-derived growth factor receptor beta (PDGFR-β) tyrosine kinases [52].

#### 3.1.1. CRISPR/Cas9 High-Throughput Screening Identifies miRNAs with a Role in Sorafenib Sensitization

A genomewide CRISPR/Cas9 library screening identified the deficiency of miR-15a (belonging to the tumor suppressor miR-15a/16-1 miRNA cluster) and miR-20b (belonging to the oncogenic miR-19~92 miRNA cluster) as contributing to sorafenib resistance in the HCCLM3 cell line. In agreement with the opposite role of these two miRNA clusters in tumors, miR-15a-overexpressing cells decreased in vivo tumorigenesis, while miR-20b overexpression slightly increased tumor size; nevertheless, the overexpression of both miRNAs led to the inhibition of tumorigenesis in the xenograft model subjected to sorafenib treatment, confirming their role in drug sensitization. Target prediction algorithms identified the cochaperone CDC37L1 (cell division cycle 37 like 1) as the only common target of these two miRNAs. Functional analysis and luciferase assays proved its inhibition by both miR-15a and miR-20b. Mechanistically, CDC37L1 binds to the heat shock protein HSP90 to activate the PPIA (peptidyl-prolyl cis-trans isomerase A) that accelerates protein folding; higher mRNA levels are associated with poorer OS and DFS in sorafenib-treated patients [53]. Another study employed a CRISPR-based screening method in vivo by using a sorafenib-treated xenograft model to improve the translational value of preclinical data with respect to in vitro tools. The authors identified miR-3689a-3p as the most overexpressed miRNA in sorafenib-sensitive tumors and reported the targeting of CCS (copper chaperone for superoxide dismutase), which, by reducing SOD1 (superoxide dismutase 1)’s ability to scavenge mitochondrial ROS, increased the cellular oxidative stress that eventually mediated the antitumor effect of sorafenib. Orthotopic mouse models showed that miR-3689a-3p downregulation decreased sorafenib efficacy. Since lower miR-3689a-3p levels were detected in tumor specimens from HCC patient cohorts, the study paves the way towards a combined miRNA mimic and sorafenib strategy to boost sorafenib’s anticancer efficacy [54]. Notably, this high-throughput screening procedure is particularly suitable for the discovery of driver genes and therapeutic targets that modulate drug efficacy. An example is represented by the metabolic gene PHGDH (phosphoglycerate dehydrogenase), regulating the serine synthesis pathway, whose specific inhibition by NCT-503 acted synergistically with sorafenib to abolish in vivo tumorigenesis [55]. Similarly, the downregulation of Kelch-like ECH-associated protein 1 (KEAP1) in response to sorafenib administration increased the activity of Nrf2, a key transcription factor controlling antioxidant responses, which contributed to enhance drug resistance to sorafenib, lenvatinib, and regorafenib in HCC [56].

#### 3.1.2. Noncoding RNAs Affect Sorafenib Response by Modulating Hypoxia-Related Signaling and Angiogenesis

Due to the antiangiogenic properties of sorafenib, blocking factors having a mitogenic effect on endothelial cells and interfering with HIF-1A (hypoxia-inducible factor 1 alpha) signaling represent crucial actions to potentiate sorafenib efficacy and to prevent drug resistance. MicroRNA-494 is an oncogenic HCC-associated miRNA which is upregulated in 30% of cases and associates with stem cell-like characteristics and poor prognosis [57,58]. Regarding its involvement in sorafenib resistance, we previously demonstrated that miR-494 activates the AKT/mTOR pathway by targeting PTEN and reported a stronger antitumor effect of antagomiR-494 plus sorafenib treatment with respect to sorafenib alone in the DEN-HCC rat model [59]. Notably, GOLPH3 (Golgi phosphoprotein) is involved in sorafenib resistance in vivo by increasing the microvascular density of xenograft tumors. Gao et al. showed that exosomes released by GOLPH3-overexpressing HCC cells are enriched in miR-494 content, enhancing tube formation and migration in the umbilical endothelial HUVEC cell line. MiR-494-loaded extracellular vesicles increased sorafenib resistance in HCC cells, highlighting both autologous and heterologous mechanisms of action for this miRNA [60]. These data proved the biologic activity of exosome-associated miR-494 in cell-to-cell crosstalk and confirmed that miR-494 is an important tumor-derived autocrine and paracrine signal, promoting angiogenesis, HIF-1A activation, and tumor growth under hypoxic conditions in different cancer types [61]. Similarly, the polypeptide 14-3-3η is a growth-promoting factor highly expressed in tumor and vascular endothelial cells, contributing to poor survival of HCC patients [62]. The study by Shen and coworkers described the overexpression of 14-3-3η in sorafenib-resistant (SR) Huh-7 cells and demonstrated that its silencing restores drug sensitivity and reduces cancer stem cell (CSC) properties. Interestingly, 14-3-3η polypeptide post-transcriptionally activates HIF-1A via inhibition of the proteasome machinery. MiR-16 was identified as the epigenetic regulator of this polypeptide, showing an inverse correlation in HCC patients treated with combined transarterial chemoembolization (TACE) and sorafenib. In this line, the low miR-16/high 14-3-3η HCC subgroup showed the worst OS after combined treatment. MiR-16 overexpression or 14-3-3η silencing in combination with sorafenib determined a higher antitumor effect in xenograft mice with respect to sorafenib alone, highlighting miR-16 restoration as a promising strategy to improve sorafenib efficacy in HCC [63]. 

#### 3.1.3. Noncoding RNAs Affect Sorafenib Response by Interfering with Tumor Cell Metabolism

Metabolic reprogramming from oxidative phosphorylation to aerobic glycolysis, also known as the “Warburg effect”, is a core hallmark of cancer cells [64] influencing the response to sorafenib. Even though ATP production during aerobic glycolysis is much lower, the Warburg effect confers advantages to cancer cell growth by providing the carbon sources required for rapid cell proliferation and, in the meantime, by minimizing the production of toxic ROS [65]. Several HCC-specific miRNAs are involved in this glycolytic shift, as in the case of miR-3662, which is downregulated in liver tumors. Its reinforced expression in HCC cell lines is associated with a decrease in glucose and oxygen consumption, ATP and lactate production, and in vivo tumorigenesis. Interestingly, both HIF-1A and HK2 (hesokinase 2) are direct targets of this miRNA, and their overexpression mitigates the above-mentioned effects, confirming their key activity in mediating miR-3662 biologic processes [66,67]. We recently reported that oncomiR-494 can rewire the tumor metabolism of HCC cells by targeting the catalytic subunit of G6pc (Glucose-6 phosphatase), which is a multi-subunit complex catalyzing the dephosphorylation of G6P to free glucose, playing a central role in glucose homeostasis. A negative correlation was displayed between miR-494 and G6pc in HCC patient cohorts, where lower G6pc levels were associated with high tumor grade, microvascular invasion (MVI), and larger tumor size. We demonstrated that the miR-494/G6pc axis contributes to the metabolic plasticity of cancer cells, favoring the accumulation of glycogen and lipid droplets that are exploited in the case of critical metabolic conditions (e.g., glucose deprivation), giving an advantage to the uncontrolled proliferation of malignant cells. We also showed that the miR-494/G6pc axis promotes sorafenib resistance and proposed combining antagomiR-based treatments with sorafenib or 2-deoxyglucose (2-DG) for HCC patients who may develop sorafenib resistance and who are ineligible for immunotherapy [68]. An interesting study by Zhang et al. [69] demonstrated the pivotal role of the miR-30a-5p/CLCF1 axis in modulating the metabolic shift toward the aerobic glycolysis of SR HepG2 cells and xenograft tumors. Specifically, they found a time-dependent decrease in miR-30a-5p in sorafenib-treated cells and demonstrated by functional analysis the direct targeting of the pro-inflammatory cytokine CLCF1 (cardiotrophin-like cytokine factor 1) that activates the downstream PI3K/AKT pathway controlling the proliferation and metabolic reprogramming of cancer cells. Indeed, the treatment with the AKT inhibitor MK-2206 reverted the glycolytic phenotype of SR HepG2 cells, decreasing ATP and lactate production as well as mRNA expression of the metabolic genes GLUT3 (glucose transporter 3), HK2, and PDK1 (pyruvate dehydrogenase kinase 1). Strikingly, the authors proved the therapeutic efficacy of a lipid formulation containing a chemically modified oligonucleotide (2′-O-methyl-modified miRNA conjugated with cholesterol) that mimics miR-30a-5p, which was injected into the tail vein of immunocompromised mice (once a week for five weeks). This miRNA formulation effectively inhibited the tumor growth of SR HepG2 cells, proving the feasibility and safety of miRNA delivery in vivo and its efficacy against sorafenib-resistant tumors. An inverse correlation between miR-30a-5p and CLCF1 was found in HCC patients confirming the importance of this signaling axis in human tumors and suggesting the combination of agomiR-30a-5p and sorafenib as a promising strategy to improve TKI efficacy and overcome acquired resistance. 

#### 3.1.4. Noncoding RNAs Affect Sorafenib Response by Interfering with Ferroptosis

From a treatment perspective, small extracellular vesicles (EVs), which are loaded with miRNAs, proteins, and mRNAs, protecting them from degradation, represent promising drug delivery vehicles and ideal miRNA carriers to cancer cells [70]. Mesenchymal stem cells are a precious source of EVs, retaining the characteristics of their parental cells and showing low immunogenicity and tumor-delivery properties [71]. An elegant study by Sun et al. described the engineering of EVs with miR-654-5p by in vitro electroporation (m654-sEV) and reported their effectiveness in sorafenib sensitization in preclinical models derived from SR-resistant HCC cells through the direct targeting of the ferroptosis inhibitor HSPB1 (heat shock factor-binding protein 1) [72]. In vivo findings proved that the combination of sorafenib and m654-sEV strongly suppressed tumor growth in comparison to sorafenib treatment alone by modulating ferroptosis-associated markers. In particular, the combined treatment effectively inhibited HSPB1 expression, increased levels of TFRC (transferrin receptor), COX2 (cyclooxygenase 2), Fe^2+^, and ROS, together with a decrease in glutathione (GSH) levels, suggesting this strategy as a reliable one to overcome sorafenib resistance in HCC. Another miRNA involved in sorafenib resistance via the impairment of iron-associated programmed cell death, ferroptosis, is miR-23a-3p, which is overexpressed in sorafenib-resistant patients and associated with tumor recurrence. A sorafenib-resistant xenograft model obtained by inoculation of MHCC97L cells showed that miR-23a-3p is overexpressed in tumors acquiring resistance after long drug exposure. Tumor-derived resistant cell lines displayed a transcriptional activation of pri-miR-23a mediated by ETS1 transcription factor. A consistent reduction in cell growth was obtained in an orthotopic model when miR-23a-3p knockout HCC cells where injected in the presence of sorafenib treatment. A functional analysis assessed the targeting of Acyl-CoA synthetase long-chain family member 4 (ACSL4), a necessary enzyme for catalyzing lipid peroxidation during ferroptosis, suggesting the silencing of miR-23a-3p as a promising option in sorafenib-resistant HCC patients [73]. 

#### 3.1.5. Noncoding RNAs Affect Sorafenib Response by Activating Oncogenic Pathways

The reactivation of oncogenic pathways is a common mechanism of drug resistance to TKIs in HCC [74], and PI3K/AKT alterations might predict sorafenib resistance [10]. We described the dual role of miR-30e-3p, which is progressively downregulated from a normal liver to a cirrhotic liver to HCC, on tumorigenesis and sorafenib resistance based on TP53 status [37]. We showed that miR-30e-3p behaves as a TS miRNA in p53 wild-type cells, establishing a feedforward loop with the TP53/MDM2 axis while it behaves as an oncogene in p53-mutated backgrounds, targeting the PTEN/AKT pathway and driving drug resistance. In a DEN-HCC rat model treated with sorafenib, which highly mirrors the human disease [75], a lower miR-30e-3p expression was detected in nonresponder tumors, displaying a negative correlation between miR-30e-3p and tumor size and a positive correlation with apoptotic markers, demonstrating the involvement of this miRNA in sorafenib sensitization. Another study reported the downregulation of miR-124-3p.1 in liver tumors and described its role in sorafenib response by targeting SIRT1 (sirtuin 1) and AKT2, preventing the nuclear translocation of FOXO3a (forkhead box O3) transcription factor. The treatment combination of miR-124-3p.1 mimics and sorafenib improved the latter’s antitumor effect in a nude mouse model [76]. Several studies reported the upregulation of the IGF/FGF pathways during acquired resistance to sorafenib [77,78]. Lin and colleagues investigated the mechanisms that lead to miRNA deregulation in SR cells. They identified the downregulation of XPO5 (exportin 5) via DNA promoter methylation to be responsible for impaired miR-378a maturation, driving IGF1R (insulin growth factor receptor 1) signaling activation [79]. The anticancer strategy pursued by the authors took advantage of GW3965, an agonist molecule of the transcription factor LXRα, which mediates miR-378a transcription. Sorafenib plus GW3965 therapy demonstrated a consistent inhibition of tumor growth compared with sorafenib alone in both orthotopic and (PDX) mouse models, demonstrating the regulation of miRNA biogenesis as a promising option to improve sorafenib effectiveness in HCC. FGFR4 (fibroblast growth factor receptor 4) and EGFR (epithelial growth factor receptor) oncogenes are upregulated in SR-resistant cell lines and are direct targets of miR-486-3p, which is downregulated in liver tumors and correlates with poor survival [80]. Intratumor injection of lentiviral particles carrying miR-486-3p in sorafenib-resistant SK-Hep1-derived orthotopic mice synergistically improved sorafenib efficacy. Similarly, a circular RNA named circRNA-SORE was shown to be upregulated in SR HCC cells and xenograft and PDX models due to increased N6-methyladenosine (m6A) levels that positively influenced its mRNA stability. Lower circRNA-SORE was associated with better OS and recurrence-free survival in sorafenib-treated patients. It acted as a ceRNA by sequestering miR-103a-2-5p and miR-660-3p, promoting the Wnt/β-catenin pathway activation that triggers and maintains a drug-resistant phenotype. Orthotopic models with sorafenib-resistant SK-Hep1 cells silenced for circRNA-SORE displayed a higher sensitization to sorafenib treatment. In agreement, intratumor injection of short hairpin lentiviral particles in sorafenib-resistant HCCLM3-derived xenograft mice potentiated the antitumor effect of sorafenib, suggesting the clinical potential of ncRNA-based combined strategies [81]. The only concern relative to the last two studies regards the use of the SK-Hep1 cell line for the establishment of orthotopic animal models. Indeed, SK-Hep1 cells originate from liver endothelial cells and not from parenchymal tumor hepatocytes. The use of inappropriate animal models may be one of the causes affecting preclinical data translation into the clinical practice; therefore, particular attention should be paid when choosing preclinical tools. Circular RNA cDCBLD2 was upregulated in SR cell lines, where it sponged miR-345-5p, increasing TOP2A expression (type IIA topoisomerase), which reduced the sorafenib-mediated apoptotic effect. Higher TOP2A expression was associated with recurrence and metastasis in HCC patients treated with sorafenib and with worse OS and recurrence-free survival. Local injection of cholesterol-conjugated small interfering RNA molecules in a sorafenib-resistant PDX model increased drug sensitivity, supporting the clinical potential of cDCBLD2 silencing to enhance sorafenib efficacy in resistant patients [82].

#### 3.1.6. Noncoding RNAs Affect Sorafenib Response by Modulating Autophagy

Regarding the role played by autophagy on sorafenib resistance, Li and coworkers reported the nuclear activation of the lncRNA SNHG1 by miR-21 in SR HCC cells and described the activation of the AKT pathway via SLC3A2 (solute carrier family 3 member 2) upregulation [83]. Interestingly, in vitro inhibition of this lncRNA by an anti-SNHG1 siRNA strategy induced sorafenib sensitization through the activation of autophagy and apoptotic cascade; moreover, it showed tumor inhibition in vivo by exerting a synergistic effect with sorafenib coadministration. On the contrary, miR-541 sensitized HCC cells to sorafenib treatment by inhibiting the expression of two autophagy-related genes, ATG2A (autophagy-related 2A) and RAB1B, highlighting the opposite role attributed to autophagy on drug sensitization [84]. MiR-541 is downregulated in HCC, and its low expression correlates with shorter OS and a high recurrence rate and predicts sorafenib resistance. Notably, intratumor injection of Adenoviral-miR-541 potentiated the effects of sorafenib in xenograft mice, resulting in maximal tumor growth inhibition [85]. 

#### 3.1.7. Noncoding RNAs Affect Sorafenib Response by Modulating Its Metabolism and Extrusion

Considering the central activity of the liver cytochrome P450 family in drug metabolism, He et al. investigated the effect of CYP3A4 (cytochrome P450 family 3 subfamily A member 4) on sorafenib metabolism and clearance and showed that a higher expression associates with poor survival in sorafenib-treated patients [86]. The authors demonstrated CYP3A4 targeting by miR-4277 and reported an addictive antitumor effect of miRNA mimics plus sorafenib in immunocompromised mice. In this regard, the study by Li and colleagues dissected the mechanisms downstream of the decreased expression of miR-138-1-3p in HCC and found the serine/threonine kinase PAK5 (p21 activated kinase 5) among its targets. PAK5 upregulation triggered β-catenin phosphorylation, causing its nuclear translocation which, in turn, activated the transcription of the multidrug-resistant transporter ABCB1 (ATP-binding cassette subfamily B member 1), which is responsible for sorafenib efflux and decreased effect. Notably, combined treatments with lentiviral vectors for miR-138-1-3p or PAK5 shRNA together with sorafenib had an enhanced anticancer effect with respect to sorafenib monotherapy in SR HepG2-derived xenograft mice [87].

In conclusion, ncRNAs regulate sorafenib resistance through a variety of molecular mechanisms (Figure 1). Preclinical studies demonstrated that combined strategies designed to restore the deregulated expression of ncRNAs enhance sorafenib sensitization in HCC, opening the path towards the design of focused clinical trials to improve treatment efficacy and patient survival.

### 3.2. Noncoding RNAs and lenvatinib Combination Improves the Therapeutic Response

Lenvatinib is an oral TKI targeting VEGFR, FGFR, PDGFRa, RET, and KIT [88]. A randomized phase III clinical trial demonstrated that lenvatinib is noninferior to sorafenib, showing an OS of 13.6 months; therefore, it was granted approval as a first-line treatment in 2018 [4]. However, only a low percentage of advanced HCC patients benefit from lenvatinib, with the great majority being nonresponders or developing drug resistance before or during treatment [89]. For this reason, the knowledge of the underlying molecular mechanisms and the discovery of new target genes for combination strategies are urgent clinical needs for improving lenvatinib efficacy.

#### 3.2.1. MicroRNAs Affect Lenvatinib Response by Modulating Oncogenic Pathways

Wei et al. showed that miR-3154 influences lenvatinib response, being upregulated in lenvatinib-resistant (LR) HCC cells. MiR-3154 was silenced in HCC cells treated with lenvatinib, resulting in reduced cancer stem cell markers, colony formation, and increased apoptosis. These effects were confirmed in PDX mouse models, where tumor volume was reduced upon lenvatinib treatment in low miR-3154 tumors only. Mechanistically, miR-3154 targets the transcription factor HNF4α (hepatocyte nuclear factor 4 alpha), which is indispensable for hepatocyte differentiation and critical for maintaining liver health, preventing its nuclear translocation. Moreover, in a cohort of HCC patients receiving lenvatinib after surgical resection, patients with low miR-3154 levels had a better survival compared to those bearing high levels. Considering that low miRNA levels correlate with a better response, the preliminary evaluation of miR-3154 in HCC tissue could help identify in advance those patients who may benefit from lenvatinib before treatment start. On the other hand, if confirmed by other studies, miR-3154 could represent a therapeutic target for improving lenvatinib sensitivity in HCC [90]. MiR-183-5p.1 promotes the expansion of liver tumor-initiating cells (T-ICs) by regulating the expression of MUC15 (Mucin 15), a membrane-associated mucin whose downregulation was previously associated with advanced-stage, poorly differentiated, and metastatic liver cancers [91]. Han et al. showed that downregulation of MUC15 elevated the expression of T-IC-associated markers, promoting malignant transformation of hepatocytes and spheroid formation in vitro. Consistently, downregulation or deletion of MUC15 in murine models dramatically increased tumor number, size, and liver-to-body weight ratio. These effects were mediated by increased levels of c-MET (mesenchymal–epithelial transition factor), PI3K (phosphatidylinositol 3-kinase), and p-AKT, revealing the existence of a miR-183-5p.1/MUC15/c-MET/PI3K/AKT/SOX2 (SRY-box transcription factor 2) regulatory circuit in liver T-ICs. In line with these results, miR-183-5p was upregulated in HCCs compared with normal tissues. On the contrary, HCC patients with high MUC15 expression displayed a prolonged survival following lenvatinib treatment, suggesting its evaluation as a predictor of lenvatinib response. In agreement, (PDO) and PDX models expressing low MUC15 levels were resistant to lenvatinib treatment. Given this evidence, the administration of anti-miR-183-5p.1 could be a possible strategy to increase MUC15 levels in HCC patients treated with lenvatinib [92]. Another miRNA involved in the regulation of oncogenic pathways is miR-128-3, which mediates the lenvatinib-resistance response in HCC cells by downregulating c-Met [93]. In LR HCC cells, miR-128-3p mimics strengthened the antiproliferative effects of lenvatinib by directly targeting c-Met, resulting in the downregulation of the ERK/cyclin D1 pathway, which is involved in cell cycle progression. In addition, miR-128-3p mimics enhanced lenvatinib-induced apoptosis in LR-HCC cells through the downregulation of p-Akt and p-GSK-3β (glycogen synthase kinase 3 beta) and the increase in caspase-9 and -3 cleavage. In xenograft mice injected with LR HCC cells, both lenvatinib treatment and miR-128-3p mimics resulted in significantly smaller tumors compared to controls. Notably, the combination therapy led to even smaller tumors than each monotherapy, showing higher apoptosis and lower proliferation indexes together with reduced p-Akt and p-GSK-3β expression. These findings suggest that the combined therapy of lenvatinib plus miR-128-3p mimics could be explored in clinical trials to further increase the efficacy of lenvatinib and possibly overcome the development of resistance. 

#### 3.2.2. Circular RNAs and Long Noncoding RNAs Affect Lenvatinib Response by Modulating Oncogenic Pathways

Liu et al. revealed that low basal circKCNN2 levels are associated with worse prognosis and tumor recurrence in HCC patients but also, on the other side, predispose them to the stronger antitumor effect of lenvatinib via the miR-520c-3p/MBD2 axis [94]. Mechanistically, circKCNN2 sponges miR-520c-3p, avoiding its binding to MBD2 (methyl-DNA-binding domain protein 2) and thus resulting in reduced proliferation, migration, colony formation, and cell cycle progression in HCC cells and a lower tumor burden in vivo. Moreover, cells and PDOs with lower intrinsic circKCNN2 levels were more sensitive to lenvatinib treatment but had a higher risk of tumor recurrence. On the contrary, ectopic expression of circKCNN2 together with lenvatinib treatment showed synergistic effects, possibly because they both downregulate the FGF19/FGFR4/FRS2 pathway. Indeed, circKCNN2 represses FGFR4 through the miR-520c-3p/MBD2 axis. In turn, intrinsic high levels of circKCNN2 may reduce the effectiveness of lenvatinib because the FGF19/FGFR4/FRS2 pathway is already inhibited. Conclusively, this work revealed that circKCNN2 may be a promising predictive biomarker of HCC recurrence and treatment sensitivity, as well as a therapeutic agent in combination with lenvatinib in low-expressing patients, even though caution should be paid due to its dual role in drug sensitivity and tumor recurrence. 

MT1JP is an lncRNA acting as a ceRNA for miR-24-3p. Yu et al. found the upregulation of MT1JP in LR HCC cells and showed that lenvatinib itself promotes MT1JP expression in vitro. Conversely, viability and apoptosis assays showed that the overexpression of miR-24-3p sensitizes HCC cells to lenvatinib. To better understand the molecular mechanisms governing the MT1JP/miR-24-3p-mediated lenvatinib response, the authors demonstrated that the antiapoptotic factor BCL2L2 (BCL2 like 2) is a miR-24-3p target gene, and its expression confers a survival advantage to lenvatinib-treated cells. In PDXs treated with lenvatinib, responder tumors had low MT1JP and high miR-24-3p levels, together with increased apoptotic markers, compared to nonresponders. Moreover, injection of MT1JP-overexpressing SMMC-7721 cells gave rise to bigger tumors in lenvatinib-treated xenograft mice. These data suggested that MTJ1P silencing or miR-24-3p mimics could be used as cotreatments to increase lenvatinib efficacy in HCC [95]. Another lncRNA upregulated in LR HCC cells, PDOs, and patients is LINC01607. Notably, a significant reduction in ROS production was found in an orthotopic HCC model following LINC01607 overexpression, thus being responsible for in vivo lenvatinib resistance. Mitophagy was activated following lenvatinib treatment, suggesting its contribution to the enhanced antioxidant capacity of LR HCC cells, helping them to maintain low oxidative stress levels. To explain the molecular mechanisms underlying LINC01607 deregulation, it emerged that it acts as a ceRNA for miR-892b, increasing p62-associated mitophagy. P62 also regulates Nrf2 expression, which in turn protects cancer cells from oxidative stress, regulating the expression of several antioxidant genes. Finally, a xenograft model with LR Hep3B cells silenced for LINC01607 demonstrated its synergistic effect with lenvatinib treatment, and the same was confirmed in the PDO model. Taken together, these results indicated that LINC01607 promotes the antioxidant capacity of LR HCC cells through the miR-892b/p62/Nrf2 axis [96], suggesting noncoding RNAs as promising therapeutic targets to overcome lenvatinib resistance in HCC.

#### 3.2.3. MicroRNAs Exert Antitumor Effects Comparable to Lenvatinib Treatment

Given the central role of miRNAs in regulating HCC progression and response to treatment, they were also explored as therapies on their own. MiR-22 represents an example; indeed, its reduced expression was linked to poor survival outcome in patients with HCC. Hu et al. proved that miR-22 gene therapy is an effective treatment in two orthotopic HCC mouse models, ensuring prolonged survival compared to lenvatinib without causing detectable toxicity [97]. The anti-HCC effects of miR-22 were mediated by its immunomodulatory functions in T cells. Indeed, miR-22 silenced HIF-1A and increased retinoic acid signaling in both hepatocytes and T cells, therefore repressing IL-17 (interleukine-17) pro-inflammatory signaling and inhibiting Th17 (T-helper 17) and Treg (T regulatory) cells’ expansion, while enhancing cytotoxic CD8+ T cells’ recruitment, activation, and survival. Additionally, miR-22 treatment improved metabolism, inhibited inflammation, and reduced hypoxia signaling. These data suggest miR-22 gene therapy as a novel effective option for HCC treatment that may also empower the effect of immunotherapy by favoring a cytotoxic immune response against HCC. There is evidence for using lenvatinib as a second-line therapy for HCC patients undergoing sorafenib resistance. In this context, Shi et al. underlined the importance of considering lenvatinib’s influence on miRNA expression profiles in SR Huh-7 cells, thus identifying possible targets influencing HCC cells’ sensitivity to TKIs. For instance, lenvatinib treatment reduced the expression of two HCC-associated miRNAs (miR-130b and miR-106b) whose high levels were associated with reduced OS in HCC patients. For this reason, studying this aspect and its molecular implications could be of great significance to improve HCC management in sorafenib-resistant patients [98]. 

In summary, despite the low number of studies reporting the mechanisms underlying lenvatinib resistance, here we showed that a variety of ncRNAs modulate lenvatinib response in HCC preclinical models (Figure 2), deserving attention as promising candidates for combined treatment strategies. 

## 4. Combination of ncRNA-Based Strategies and ICIs Improves Therapeutic Efficacy in HCC Preclinical Models

Immune checkpoint inhibitors improved the survival of advanced HCCs. However, certain etiologies (e.g., NAFLD) or genetic backgrounds (e.g., β-catenin mutations) may affect ICI therapeutic response in a substantial proportion of patients [9,10]. Additionally, therapeutic efficacy could be further improved in responders. An entangled network of interactions exists between tumor cells and cellular subpopulations belonging to the tumor microenvironment (TME). HCC cells can reprogram the expression pattern of several ncRNAs as follows: (I) alter cancer cell immunogenicity by decreasing the exposure of cancer-associated antigens, (II) promote the immune exhaustion of CD8+ T cells by regulating the expression of immune checkpoint inhibitors, and (III) decrease the tumor infiltration of immune system cells such as CD8+ T cells, natural killer (NK) cells, dendritic cells (DCs), and macrophages by modulating the release of pro-inflammatory and anti-inflammatory cytokines. The specific deregulation of ncRNAs in immune cells also contributes to impaired antitumor immunity. 

### 4.1. MiRNAs Modulate Gene Expression in Immune System Cells, Preventing Tumor Development

The peculiar ability of miRNAs to fine-tune gene expression makes them ideal candidates to precisely modulate immune system components. This is the case of miR-206, whose expression is reduced not only in neoplastic hepatocytes but also in Kupffer cells (KCs), driving M2 polarization and the depletion of cytotoxic T cells (CTLs) in AKT/Ras mice and leading to early tumor lethality. The specific overexpression of miR-206 in KCs promoted M1 polarization by targeting Kruppel-like factor 4 (Klf4) and, thereby, enhancing the production of M1 markers such as C-C motif chemokine ligand 2 (CCL2) that favor hepatic recruitment of CTLs. MiR-206 prevented HCC onset in mice with the AKT/Ras genetic background, suggesting its potential use as an immunotherapeutic target [99]. Similarly, hydrodynamic injection of microRNA-15a/16-1 prevented HCC in 100% of AKT/Ras and c-Myc mice and, when used in therapeutic settings, promoted tumor regression in both animal models. Mechanistically, microRNA-15a/16-1 inhibited the transcriptional activation of the chemokine CCL22 by nuclear factor-kB (Nf-kB) targeting in KCs, therefore preventing Tregs recruitment and the dysregulation of CD8+ T cells. This miRNA cluster represents a potential candidate for immunotherapy against HCC alone or in combination with current clinical agents [100].

### 4.2. Noncoding RNAs Affect Immunotherapy Response by Interfering with Tumor Cell Metabolism

Metabolic reprogramming occurring in cancer cells can induce nutrient competition with immune system cells or can lead to the accumulation of immunosuppressive metabolites that hamper antitumor immunity. Clinical trials evaluating the combined effect of metabolic modulators and immune checkpoint inhibitors are underway in many cancer types [101]. Cai and coworkers discovered a novel circular RNA upregulated in HCC, named circRHBDD1, which correlates with tumor number and size and MVI and AFP levels and predicts poor prognosis. Experimental models showed that circRHBDD1 triggers aerobic glycolysis by regulating the expression of the metabolic genes HK2 and GLUT1, thus leading to a decrease in the oxygen consumption rate with increasing lactate and ATP production and promoting in vivo tumorigenesis in a PDX mouse model. Mechanistically, circRHBDD1 interacts with the m6A-binding protein YTHDF1 (YTH N6-methyladenosine RNA-binding protein F1), favoring PIK3R1 (phosphoinositide-3-kinase regulatory subunit 1) translation and activating the PI3K/AKT pathway that contributes to the metabolic shift in cancer cells. Notably, higher levels of circRHBDD1 were detected in HCC patients with disease progression after receiving an anti-PD1 therapy. An immunocompetent xenograft model inoculated with circRHBDD1-silenced Hepa1-6 cells showed a stronger antitumor response and an improved survival in the presence of anti-PD1 treatment with respect to control cells, with a higher staining for CD8+ T cell infiltrates. These findings suggest that targeting cancer metabolism might synergistically enhance immunotherapy not only via metabolic reprogramming of cancer cells but also by reshaping the TME [102]. In this line, the dysregulation of genes belonging to fatty acid (FA) metabolism is considered an emerging cause of tumor aggressiveness and poor prognosis in several cancer types [103]. An in silico study identified an FA metabolism-related lncRNA signature which categorized patients from two online datasets on the basis of a high or low FA metabolic score. Notably, patients with a lower FA metabolic score presented a higher immune cell infiltration score, an upregulation of critical immune checkpoint inhibitor genes, and a higher tumor immune dysfunction and exclusion (TIDE) score, suggesting that this patient subgroup may experience immune escape and may have a lower probability of benefit from immunotherapy [104]. Similarly, a signature based on the expression of the lncRNA SNHG1 and on its target genes FANCD2 (FA complementation group D2) and G6PD (glucose 6 phosphate dehydrogenase), the expression of which is increased due to miR-199a sponging activity, displayed an association with poorer OS in patients within the high-risk group. These patients also showed the upregulation of multiple checkpoint molecules and a higher TIDE score, suggesting that this signature might predict an immune-suppressive-tumor milieu, representing a potential marker for the decision making of tailored therapeutic strategies in HCC [105]. 

### 4.3. Noncoding RNAs Affect Immunotherapy Response by Interfering with CD8+ T Cells Recruitment

An interesting study by Huang et al. identified the overexpression of circular RNA circMET in human HCCs and highlighted its correlation with tumor progression and shorter OS as well as with EMT in HCC cells. Mechanistically, the ceRNA activity of circMET with respect to miR-30-5p family members drove the upregulation of the downstream target Snail that, in turn, transcriptionally activated DPP4 (dipeptidyl peptidase 4), whose expression deeply impacts insulin and glucose metabolism and immune cell regulation. Regarding the latter aspect, immunocompetent animal models implanted with circMET Hepa1-6 cells showed lower CXCL10 (C-X-C motif chemokine ligand 10) serum levels and CD8+ T cell tumor infiltration. Strikingly, the combination of anti-PD1 inhibitor with the anti-DPP4 inhibitor sitagliptin in the xenograft model induced a complete tumor regression, emphasizing the improved therapeutic efficacy of ICIs when combined with inhibitors of genes targeted by HCC-specific ncRNAs [106]. Regarding the modulation of the DPP4 gene, Zhang et al. identified an oncogenic lncRNA LINC01132 which is overexpressed in human HCCs due to copy number amplification. Specifically, LINC01132 binds to NRF1 (nuclear respiratory factor 1) transcription factor, which activates the DPP4 promoter, mediating its upregulation in HCC cells. LINC01132 silencing determined tumor growth inhibition in vivo in two HCC animal models (xenograft and PDX). Due to the well-known immune suppressive activity of DPP4 [107] and its association with decreased lymphocyte trafficking, a combination therapy of LINC01132 shRNAs and PD-L1 inhibitor was tested in the Hepa1–6 xenograft model, showing a clear tumor regression with respect to LINC01132 silencing alone and the highest positivity for CD8+ infiltrates, proving the efficacy of LINC01132 knockdown together with a PD-L1 blockade [108]. 

### 4.4. Noncoding RNAs Affect Immunotherapy Response by Mediating Cell–Cell Interactions

An outstanding study by Fu et al. demonstrated the relevance of the myeloid-associated miRNA miR-223 in cell-to-cell crosstalk, modulating the tumor hypoxia, angiogenesis, and inflammatory tumor microenvironment that control HCC progression. The authors proposed an intriguing model for chronic inflammation-associated HCCs in which myeloid cells represent the source for miR-223 transfer to cancer cells, where miR-223 inhibits HIF-1A expression and indirectly influences the composition of the TME by suppressing the HIF-1A-driven CD39/CD73-adenosine pathway that contributes to PD-1 and PD-L1 upregulation in immune cells. Adenovirus-mediated gene delivery of miR-223 in two inflammatory-associated models of HCC hindered tumor development and progression by inhibiting angiogenesis and hypoxia-mediated PD1/PD-L1 activation in T cells and macrophages, proving the therapeutic potential of miR-223 in blocking the immunosuppressive tumor microenvironment in HCC [109]. Notably, “RNA–RNA” crosstalk relies not only on noncoding RNAs functioning as “microRNA sponges” but also on coding mRNAs, which can relieve the inhibitory effect of miRNAs on their target genes by ceRNA activity. An example is represented by the HMGB1 (high-mobility group box 1) mRNA, whose overexpression in HBV+ early-stage HCCs acts as a miRNA sponge to competitively bind the miR-200 family (miR-200a/200b/429), leading to RICTOR (RPTOR-independent companion of MTOR complex 2) mRNA upregulation which, in turn, activates the AKT/mTORC1 pathway. This epigenetic crosstalk leads to increased glutamine metabolism and the release of PD-L1+ exosomes that affect immunotherapy response. In this context, HMGB1 is a new therapeutic target and biomarker of anti-PD-L1 efficacy in early-HCC patients [110]. Other studies reported the incorporation of circRNAs into HCC-derived exosomes to be delivered to immune cell subpopulations to promote their dysfunction. Hu and colleagues found the upregulation of circCCAR1 in tumor tissues and the exosomal fraction from HCC, which correlated with poor prognosis. In vivo experiments revealed an increase in tumor growth and metastasis of circCCAR1-OE HCCLM3 cells. Notably, this circRNA increased the expression of WTAP (Wilms tumor 1-associated protein) by sponging miR-127-5p that, in turn, mediated m6A modification and enhanced the stability of circCCAR1 itself. The extracellular secretion of circCCAR1 into exosomes determined CD8+ T cells’ dysfunction due to direct PD1 stabilization. Experiments in humanized NOD/SCID gamma (HuNSG) mice confirmed the decrease in CD8+ T cell infiltration in tumors from circCCAR1-OE HCCLM3 cells and showed immune resistance to anti-PD1 therapy (Opdivo) with a decreased survival. In agreement, tissue and exosomal circCCAR1 levels were negatively related to CD8+ T cells in HCC patients [111]. Similarly, a higher expression of circUHRF1 was reported in HCC tissues, associating them with poor prognosis and NK cell dysfunction. Indeed, exosomal embedding and extrusion of circUHRF1 inhibited NK population, decreasing the secretion of pro-inflammatory cytokines IFN-γ and TNF-α. Mechanistically, circUHRF1 sponged miR-449c-5p, determining the upregulating of TIM-3 expression, which promoted NK cell exhaustion. A xenograft model with circUHRF1-knockdown HCCLM3 cells was established, and NK cells were injected after tumor growth. Anti-PD1 sensitization and improved OS were observed in circUHRF1-knockdown mice, highlighting that circUHRF1 inhibition might be a promising strategy to ameliorate anti-PD1 efficacy in HCC [112]. 

Dong and coworkers demonstrated that the oncogenic miR-93-5p is deregulated early during tumorigenesis. Its overexpression caused changes towards the mesenchymal phenotype and malignant transformation of liver progenitor cells (LPCs), giving rise to tumors and metastasis in 100% of cases when inoculated in animal models. A proteomic analysis revealed GAL-9 (galectin 9) upregulation in miR-93-OE LPCs, suggesting its inhibition in a therapeutic perspective. Indeed, it functions as a negative regulator of the innate response favoring antitumor immunity evasion of cancer cells. Although anti-PD1 monoclonal antibodies had no effect against miR-93-OE-derived xenografts, tumor shrinkage was observed when combining anti-GAL-9 and anti-PD1 treatments, providing evidence for promising GAL-9 targeting in combined strategies for the treatment of LPC-like HCC subtypes [113]. 

In summary, the deregulation of several noncoding RNAs influences the relationship between tumor and immune cells, often promoting the adoption of elusive mechanisms to escape from innate and adaptative immune response, allowing cancer cells to expand and invade distant sites. Table 1 summarizes the therapeutic strategies adopted by different preclinical studies to prove the efficacy of novel candidates as immunotherapeutic agents alone or in combination with ICIs for the treatment of HCC.

## 5. Noncoding RNAs As Biomarkers of Treatment Response in HCC

Noncoding RNAs are dysregulated in many cancer types including HCC and show promise as treatment response biomarkers at the tissue level [114]. In 2008, an outstanding study demonstrated for the first time the presence of miRNAs in body fluids such as serum and plasma [115]. MiRNAs can be secreted outside the cell through active or passive extrusion mechanisms, being incorporated into microvesicles or exosomes that protect them from RNase activity. Moreover, the intrinsic nature of miRNAs makes them robust candidates in the case of repeated freeze–thaw cycles of biologic specimens. The ease of miRNA detection methods (e.g., real-time PCR and digital PCR) [116] allows for their analysis in liquid biopsy, a noninvasive procedure that consists of the evaluation of repeated blood withdrawals to monitor patients during follow-up assessments. Several studies attempted to identify tissue and blood biomarkers that could be predictive of immunotherapy, but no robust results were obtained so far [117]. Although tissue may represent the preferred source, biomarker discovery could take advantage of liquid biopsy to overcome the problem of tumor heterogeneity and to avoid the hazard of liver biopsy at advanced stages.

Since sorafenib has been the only first-line therapeutic option for nearly a decade [2], showing modest survival benefits due to the onset of innate or early-acquired resistance [118], great efforts have been made to search for circulating biomarkers for an early switch to second-line agents. In this setting, our group reported the sorafenib-mediated extrusion of oncomiR-221 in preclinical models, showing an inverse correlation with tissue levels in treated animals (DEN-HCC rats and xenograft mice). To test whether miR-221 could be a predictive biomarker of sorafenib response, miR-221 levels were analyzed in the sera of two cohorts of HCC patients both before and during treatment (two-month follow-up). In line with its oncogenic properties, lower miR-221 levels were detected in responder patients at the basal level. On the contrary, comparisons between different time points showed an increase in serum miR-221 levels in responder patients when evaluated during treatment with respect to pretreatment values. We suggested that, if confirmed in future studies, monitoring miR-221 circulating levels over time in sorafenib-treated patients might represent a promising strategy to discriminate patients with prolonged response from those with early tumor escape, increasing the treatment window for second-line options for the latter ones. Moreover, we demonstrated that high miR-221 tumor levels influence sorafenib resistance due to caspase-3 inhibition, leading to decreased apoptotic cell death [119]. De la Cruz-Ojeda investigated the relationship between three sorafenib-deregulated miRNAs (miR-200c-3p, miR-222-5p, and miR-512-3p) and sorafenib response in advanced HCCs. Notably, a hazard ratio analysis showed an association between miR-200c-3p baseline levels and increased survival, while miR-222-5p and miR-512-3p levels, analyzed one month after sorafenib treatment, were associated with a poorer prognosis [120]. Since miR-221 and miR-222 belong to the same bicistronic cluster, findings between that study and ours seem discordant, pointing out that further efforts need to be put in place to obtain robust data that can be translated into laboratory tests, in terms of methodology and evaluation of different patient cohorts. Regarding the identification of early-escape biomarkers, Gramantieri et al. demonstrated that sorafenib modulates exosome-mediated miR-30e-3p extrusion in a TP53-dependent manner, promoting the increase in its circulating fraction. This finding was supported by the inverse correlation between tissue and serum miR-30e-3p levels observed in DEN-HCC rats and xenograft mice subjected to sorafenib treatment. In a preliminary cohort of sorafenib-treated HCC patients, higher circulating miR-30e-3p levels were found in the sorafenib-resistant group when evaluating samples collected after 2 months of treatment, suggesting miR-30e-3p as a possible candidate for predicting the development of sorafenib resistance [37].

The recent approval of novel molecular-targeted drugs and immunotherapy-based therapies has revolutionized the management of advanced HCC patients; however, no biomarker has entered clinical practice to support clinicians in stratifying patients not eligible for ICIs. In sorafenib-treated patients, Fernández-Tussy and collaborators reported the relationship between higher miR-518d-5p circulating levels and shorter treatment duration and OS in the BCLC-C patient subgroup only. Mechanistically, this oncogenic miRNA belonging to the C19MC family targets PUMA and confers a survival advantage to cancer cells by enhancing their buffering capacity against ROS, maintaining membrane integrity and avoiding apoptosis during sorafenib treatment [121]. Nishida et al. performed a miRNA screening in the serum of 16 HCC patients treated with sorafenib, identifying miR-181a-5p and miR-339-5p as associated with disease response at 1 month of follow-up. In a validation cohort (53 patients), miRNA levels decreased progressively among patient groups, showing high levels in partial responders, intermediate levels in patients with stable disease, and low levels in those with progressive disease. Again, when BCLC-C cases were considered, multivariate analysis at the 3-month follow-up revealed miR-181a-5p as an independent factor for predicting disease control and OS. These last two studies underline the high heterogeneity of advanced HCCs and, as expected, report a better predictive performance of circulating miRNAs within more homogeneous patient subgroups [122]. The study by Shao et al. investigated miR-10b-3p’s role in regulating tumor response to sorafenib treatment. In preclinical models, sorafenib induced miR-10b-3p extrusion, and higher miR-10b-3p levels were associated with sorafenib sensitization, leading authors to hypothesize its possible use as a biomarker for patients treated with sorafenib. In a small patient cohort, higher miR-10b-3p serum levels predicted a better OS but not progression-free survival when analyzed before treatment [123]. Interestingly, miR-10b is overexpressed in several tumors including HCC, and its expression in the exosomal fraction from early-HCC patients is closely associated with tumor size and recurrence, representing an independent prognostic factor for poor survival [124]. Since two miR-10b inhibitors have been developed based on advances in nanotechnology (TTX-MC138 and RGLS5579), demonstrating an effective anticancer activity in preclinical models [50], this miRNA investigation holds promise both as a therapeutic target and a predictive biomarker. Finally, a study by our research group reported the association between high miR-494 serum levels and sorafenib resistance in HCC patients at baseline [68]. In addition, we underlined the relationship between miR-494 serum levels and genes involved in the metabolic reprogramming of cancer cells, pointing out the possibility that circulating levels of this miRNA might not only predict sorafenib response but also identify tumors with a deregulated metabolism that could benefit from combined TKI and antagomiR or metabolic interference strategies. The preliminary study needs to be confirmed in larger patient cohorts to prove the predictive value of miR-494 for patient stratification to tailored treatments. 

Regarding lenvatinib, one study assessed the correlations between serum biomarkers and efficacy outcomes from the REFLECT clinical trial. Remarkably, only serum proteins were tested by ELISA, whereas no data on circulating ncRNAs are reported. Briefly, higher baseline VEGF, ANG2, and FGF21 correlated with shorter OS with both sorafenib and lenvatinib treatments, while a longer OS correlated with higher baseline FGF21 in the lenvatinib group compared with sorafenib, which needs further confirmation [125].

Regarding second-line agents, regorafenib is a multiple TKI, blocking the activity of protein kinases that regulate angiogenesis, proliferation, tumor microenvironment, and metastasis, including VEGFR1-3, TIE2, KIT, RET, RAF-1, BRAF, PDGFR, and FGFR. There is only one retrospective study, performed in the RESORCE trial, reporting the association between circulating miRNAs and prolonged OS in regorafenib-treated patients [126]. Plasma specimens from 343 HCC patients (234 regorafenib-treated and 109 placebo) were assessed for the expression of 750 miRNAs. Nine circulating miRNAs (increased: MIR30A, MIR122, MIR125B, MIR200A, MIR374B; decreased: MIR15B, MIR107, MIR320; and absent: MIR645) were predictive of survival benefit with regorafenib. Top gene sets related to these miRNAs included liver cancer progression and metabolic pathways such as lipids, amino acids, bile acids, and xenobiotics metabolism, and glucuronidation. Bioinformatics analysis revealed that patients with improved regorafenib response overlap with the well-differentiated S3 subtype of the Hoshida classification [127], which is characterized by a hepatocyte-like phenotype and well-differentiated and smaller tumors. Notably, AFP and c-MET plasma levels were associated with decreased overall survival independent of regorafenib treatment, while only five plasma proteins (angiopoietin 1 [ANG-1], cystatin B, the latency-associated peptide of transforming growth factor beta 1 [LAP TGF-b1], oxidized low-density lipoprotein receptor 1 [LOX-1], C-C motif chemokine ligand 3 [MIP-1a]) were associated with treatment benefit.

In summary, although the measurement of tumor-derived miRNAs in body fluids might represent an easy and promising approach for the blood-based detection of treatment response or early tumor escape in HCC (Table 2), further studies in larger patient cohorts or well-defined patient subgroups are needed to translate these findings into routine lab tests. Studies designed to validate biomarkers and to identify new ones are critical for the improvement of tailored treatments in HCC.

## 6. Future Perspectives

The results presented in this review reveal that therapeutic strategies aimed at restoring the expression of deregulated noncoding RNAs potentiate the antitumor effect of immunotherapy and molecular-targeted therapy in the preclinical setting. These encouraging findings, together with information derived from ongoing clinical trials assessing miRNA formulations, could be helpful to potentiate first-line therapeutic strategies preventing the onset of disease progression and drug resistance. Future preclinical and clinical studies should be designed to identify efficient formulations for the targeted delivery of modified oligonucleotides to liver tumors, avoiding adverse immune events and off-target effects.

The lack of circulating biomarkers still represents the Achilles’ heel if the optimization of patient management and the development of a true personalized therapeutic strategy in HCC. As we are facing a breakthrough in treatment availability for advanced patients, further efforts need to be deployed to identify biomarkers for patient stratification and early tumor escape. The identification of multiparameter signatures combining one or more noncoding RNAs and clinicopathological variables (e.g., serum AFP, albumin, bilirubin, etc.) might increase the predictive score for treatment response. Due to tumor heterogeneity, multicenter studies could guarantee a higher probability of success in the identification of predictive biomarkers.

Although ncRNAs are unlikely to be the “magic bullet” for all HCC patients, they could be important “weapons” for combination strategies in selected subgroups of patients and could be discriminating features when selecting patients for personalized treatments and sequence options.

## 7. Conclusions

Preclinical studies unraveled the molecular mechanisms underlying the deregulated noncoding RNAs associated with the drug-resistance phenotype in HCC and demonstrated their potential in combined therapeutic strategies to be evaluated in “ad hoc” clinical trials with well-defined nanoparticle delivery systems. Because of the numerous noncoding RNAs with a role in treatment response, it remains challenging to select the most promising candidates for combined therapeutic interventions. Notably, some studies reported the feasibility and efficacy of blocking downstream targets of HCC-associated ncRNAs by using inhibitors or monoclonal antibodies, increasing the range of possible therapeutic approaches for overcoming HCC drug resistance, contributing to a better outcome in advanced patients. Since liver biopsy is often not feasible in patients with advanced diseases, the use of preclinical models can provide useful information about the relationship between circulating and tissue ncRNAs, which may help in selecting patient subgroups eligible for combined treatments based on extracellular levels. Notably, the first phase 1 clinical trial testing a miRNA mimic formulation in several metastatic cancer types, including HCC, was terminated owing to serious immune-mediated adverse events [49]. The tested drug was MRX34, which is a synthetic double-stranded miR-34a mimic encapsulated in liposomal nanoparticles (ClinicalTrials.gov identifier: NCT01829971). The lesson from this study was that improvements in synthetic miRNA mimics and delivery systems are necessary. Despite this first failure, other clinical trials testing miRNA formulations in cancer are ongoing [50], pointing out the potential of miRNA therapy as a next-generation strategy.

Regarding blood biomarkers, the translation of preclinical findings is still far from being put into practice, and several causes may be responsible for this issue. First, there are too many variables that prevent a complete comparison between the different studies; to name a few, there are different samples (plasma or serum or exosomes), different analytical methodologies (qPCR, ddPCR, RNAseq), different etiologies among the human cohorts, and different time points (at baseline or on treatment). Remarkably, the majority of studies investigating circulating miRNAs as diagnostic biomarkers are from Eastern populations. Therefore we cannot rule out a role of etiology and comorbidities as possible factors affecting the performance of these putative diagnostic markers, given the heterogeneity of patient background in Western and Eastern cohorts [128,129]. In addition, because of the heterogeneity of advanced HCCs, it could be necessary to divide patients into more homogeneous subgroups (e.g., BCLC-C, etiology, driver gene mutations, etc.) where the predictive value of biomarkers might perform better. Additional clinical needs concern patients not eligible for immunotherapy, for whom the choice among sorafenib and lenvatinib remains, and patients undergoing tumor escape with first-line treatments to improve the likelihood of success with second-line agents. Finally, the identification of multiparameter signatures combining one or more noncoding RNAs and clinicopathological variables (e.g., serum AFP, albumin, bilirubin, etc.) might increase the predictive score in terms of treatment response. 

In conclusion, giant strides have been made in preclinical studies regarding the possible use of noncoding RNA-based strategies to improve the efficacy of current therapies. Results from ongoing clinical trials in cancer patients will allow a step forward in the near future, when combined approaches to overcome the onset of drug resistance in HCC patients can be tested.

## Figures and Tables

**Figure 1 cancers-16-00766-f001:**
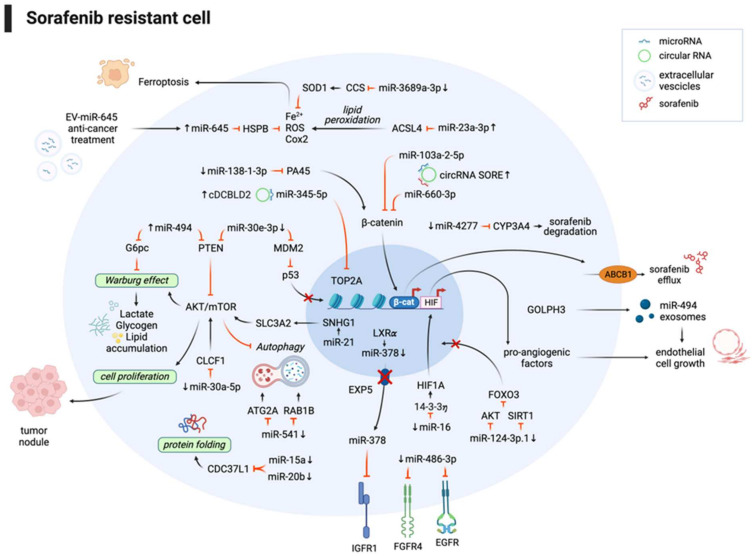
Molecular mechanisms underlying deregulated noncoding RNAs influencing sorafenib resistance in HCC. Schematic representation of molecular pathways playing a role in sorafenib-resistant cells following aberrant expression of HCC-specific ncRNAs. Black arrows connecting ncRNAs and genes mean a positive effect. Red arrows connecting ncRNAs and genes mean an inhibitory effect.

**Figure 2 cancers-16-00766-f002:**
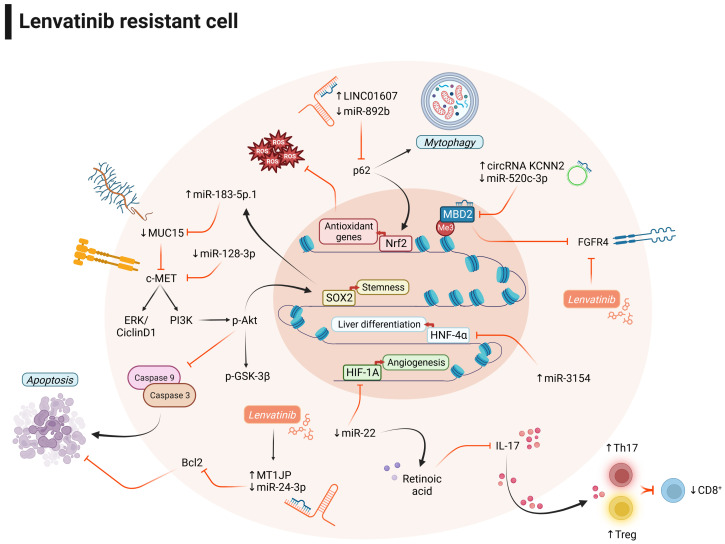
Molecular mechanisms underlying deregulated noncoding RNAs influencing lenvatinib resistance in HCC. Schematic representation of molecular pathways playing a role in lenvatinib-resistant cells following aberrant expression of HCC-specific ncRNAs. Black arrows connecting ncRNAs and genes mean a positive effect. Orange arrows connecting ncRNAs and genes mean an inhibitory effect.

**Table 1 cancers-16-00766-t001:** Experimental models for evaluating noncoding RNAs as immunotherapy targets.

Noncoding RNA	Target Gene/Sponged miRNA/Other Targets	Experimental In Vivo Models	Therapeutic/Experimental Strategy	Effect on Immune Cells	Treatment Combination	Ref. No.
miR-206	Klf4/CCL2	AKT/Ras and Sleeping Beauty transposon hydrodynamic injection in FVB/NJ mice	Minicircle and Sleeping Beauty hydrodynamic injection for miR-206 overexpression	Decreased Treg recruitment	None	[99]
miR-15a/16-1	Nf-kB/CCL22	AKT/Ras, Myc hydrodynamic injection in FVB/NJ mice	Hydrodynamic injection for miRNA overexpression	M1 macrophage polarization	None	[100]
circRHBDD1	YTHDF1/PIK3R1	PDX NOD/SCID, BALBc mice; Hepa1-6 cells in xenograft C57BL/6 mice	circRHBDD1 interference vector	N/A	Anti-PD1	[102]
circMET	miR-30-5p/SNAI1/DPP4/CXCL10	Hepa1-6 cells in xenograft C57BL/6 mice	Sitagliptin (DPP4 inhibitor)	Increased CD8+ T cells recruitment	Anti-PD1	[106]
LINC01132	NRF1/DPP4	PDX nude mice; Hepa1-6 cells in C57BL/6 xenograft mice	LINC01132 adenovirus interference vector	Increased CD8+ T cells recruitment	Anti-PD-L1	[108]
miR-223	HIF1/CD39/CD73	miR-223 KO mice + DEN or CCL4; C57BL/4J mice + DEN+CCl4	miR-223 adenovirus vector	Decreased PD1/PD-L1 expression	None	[109]
CircCCAR1	miR-127-5p/WTAP	HCCLM3 cells in BALBc, HuNSG xenograft mice	circCCAR1 overexpression vector	CD8+ T cells dysfunction	Anti-PD1	[111]
circUHRF1	miR-449c-5p/TIM3	HCCLM3 cells in NOD/SCID xenograft mice	circUHRF1 interference vector	Increased NK activity	Anti-PD1	[112]
miR-93-5p	GAL-9	LPC cells in xenograft and orthotopic nude mice	Anti-GAL-9	Increased CD8+ T cells recruitment	Anti-PD1	[113]

**Table 2 cancers-16-00766-t002:** MicroRNAs as biomarkers in advanced HCCs.

miRNA Name	Blood Specimen	Timepoint of Analysis	Circulating Levels in Responders	Treatment	Ref. No.
miR-221	Serum	Basal On treatment (2 m)	Low High	Sorafenib	[119]
miR-200c-3p miR-222-5p miR-512-3p	Plasma	Basal On treatment (1 m) On treatment (1 m)	High Low Low	Sorafenib	[120]
miR-30e-3p	Serum	On treatment (2 m)	Low	Sorafenib	[37]
miR-518d-5p	Serum	Basal	Low	Sorafenib	[121]
miR-181a-5p	Serum	Basal	High	Sorafenib	[122]
miR-10b-3p	Serum	Basal	High	Sorafenib	[123]
miR-494	Serum	Basal	Low	Sorafenib	[68]
miR-30a, miR-122, miR-125b, miR-200a, miR-347b; miR-15b, miR-107, miR-320; miR-645	Plasma	Basal	High Low Absent	Regorafenib	[126]

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
