# Peer review of "Noncoding RNAs in Hepatocellular Carcinoma: Potential Applications in Combined Therapeutic Strategies and Promising Candidates of Treatment Response"

_cancers, 2024, doi:10.3390/cancers16040766_

Round 1

Reviewer 1 Report

Comments and Suggestions for Authors

This review manuscript provided a comprehensive review of recent preclinical studies using animal models to investigate the synergistic impact of non-coding RNAs (microRNAs, long non-coding RNAs, and circular RNAs) with TKIs or ICIs. It highlights the rationale behind unconventional combination strategies. The review emphasizes non-coding RNAs, particularly microRNAs, as potential circulating indicators of treatment response and tumor escape in HCC. This reviewer has two concerns:

1.     The graph is well-crafted, but discrepancies exist between the title, introduction and the main content. The authors described “the Role of non-coding RNAs in hepatocarcinogenesis” where circular RNAs are emphasized alongside microRNAs and long non-coding RNAs. However, the below main text predominantly focuses on microRNAs, with limited discussion on circular and no mention of long non-coding RNAs. An adjustment is recommended to better reflect the manuscript's content.

 2.     Part 3 and Part 4 highlight the combination of non-coding RNAs with TKIs and ICIs in HCC. It would be helpful to understand the author's review outline for improved clarity. For enhanced clarity, please consider subdividing these parts into smaller sections to clearly delineate the content, facilitating the audience's comprehension of the manuscript's structure.

Comments on the Quality of English Language

NA

Author Response

  • Minor editing of English language required

Answer: The text has been subjected to English editing by all the Authors. The corrections are visible in the tracked version of the manuscript.

This review manuscript provided a comprehensive review of recent preclinical studies using animal models to investigate the synergistic impact of non-coding RNAs (microRNAs, long non-coding RNAs, and circular RNAs) with TKIs or ICIs. It highlights the rationale behind unconventional combination strategies. The review emphasizes non-coding RNAs, particularly microRNAs, as potential circulating indicators of treatment response and tumor escape in HCC. This reviewer has two concerns:

  1. The graph is well-crafted, but discrepancies exist between the title, introduction and the main content. The authors described “the Role of non-coding RNAs in hepatocarcinogenesis” where circular RNAs are emphasized alongside microRNAs and long non-coding RNAs. However, the below main text predominantly focuses on microRNAs, with limited discussion on circular and no mention of long non-coding RNAs. An adjustment is recommended to better reflect the manuscript's content.

Answer: As requested by the Reviewer, we added four new updated references for both the long non-coding RNAs (Ref. N. 17, 18, 22, 23) and circular RNAs (Ref. N. 25, 26, 28, 30) related chapters. Regarding lncRNAs, we added and discussed studies relative to their ability to codify for proteins, to their expression in healthy, cirrhotic and HCC specimens and to the regulation of driver genes (TP53 and CTNNB1), suggesting their possible use as therapeutic targets in molecular-stratified HCC patients. Regarding circRNAs, we added and discussed studies relative to their ability to codify for proteins, to the regulation of autophagy and AKT/mTOR pathways, to their involvement in triggering metabolic reprogramming of HCC cells as well as stem cell properties. For both paragraphs, we added a conclusive sentence on the possible use of lncRNAs and circRNAs as therapeutic candidates and biomarkers. We thank the Reviewer for this suggestion aiding to improve the quality and the focus of our manuscript. We think that the three mentioned subchapters are now more balanced.

  1. Part 3 and Part 4 highlight the combination of non-coding RNAs with TKIs and ICIs in HCC. It would be helpful to understand the author's review outline for improved clarity. For enhanced clarity, please consider subdividing these parts into smaller sections to clearly delineate the content, facilitating the audience's comprehension of the manuscript's structure.

Answer: As requested, we divided Part 3 and Part 4 in subchapters headed by specific subtitles. The order of same paragraphs has been changed to better gather studies under defined topics. We thank the Reviewer for this suggestion that improved the clarity and comprehension of our manuscript.

Reviewer 2 Report

Comments and Suggestions for Authors

This is a very interesting article and well organized, and will be acceptable for publication after minor revision.

Minor points

1)    Abbreviation: Please add a list of abbreviations. It will help read smoothly the manuscript.

2)    Manuscript structure: Please add “In summary“ or “In conclusion” in 2. Role of non-coding RNAs in hepatocarcinogenesis, and 5. Non-coding RNAs as biomarkers of treatment response in HCC. Please add “Future perspectives” before Conclusion.

3) Clinical points of view: Are there some differences of results among races? Between HCCs with/without LC.

Author Response

Minor points

1)    Abbreviation: Please add a list of abbreviations. It will help read smoothly the manuscript.

Answer: As requested, a list of abbreviations has been added to the end of the manuscript after references. The abbreviations are in order of mention. Regarding gene nomenclature, we mentioned in the text the abbreviation of the gene name followed by the full gene name (inside brackets). We thank the Reviewer for this suggestion that improved the readability of our manuscript.

2)    Manuscript structure: Please add “In summary“ or “In conclusion” in 2. Role of non-coding RNAs in hepatocarcinogenesis, and 5. Non-coding RNAs as biomarkers of treatment response in HCC. Please add “Future perspectives” before Conclusion.

Answer: “In summary” has been added to the last paragraph of chapter N.2 and 5. A new chapter named “Future perspectives” has been added before Conclusions.

3) Clinical points of view: Are there some differences of results among races? Between HCCs with/without LC.

Answer: Regarding microRNA profiling, we have found some discrepancies in miRNA signatures among Caucasian and Asian cohorts, even though we cannot exclude those discrepancies may arise from methodological aspects rather than from the geographic origin of patient cohorts. For example, only a partial overlap was found between our 35-miRNA signature (doi: 10.1158/0008-5472.CAN-06-4607) and the 30-miRNA signature identified by Murakami et al. (doi: 10.1038/sj.onc.1209283). The study by Murakami did not identify the downregulation of the liver-specific miR-122 which is deregulated in almost all liver diseases including HCC. Notably, both cohorts compared HCC tissues versus surrounding cirrhotic livers, which may account for the similarity in the deregulation of some miRNAs such as miR-199a, miR-195 and members of the let-7 family.

Moreover, the different etiology of HCC cohorts coming from different geographical areas might play an important role in the deregulation of non-coding RNAs. Several studies (doi: 10.1002/hep.22749; doi: 10.1002/ijc.28075) reported miRNA signatures differentiating HBV-related (more frequent in Asian and Sub-Saharan Africa cohorts) from HCV-related cases (more frequent in Western and Japanese cohorts, at least before the advent of DAA treatments for HCV). Remarkably, the majority of studies investigating circulating miRNAs as diagnostic biomarkers are from eastern populations. Therefore we cannot rule out a role of etiology and comorbidities as possible factors affecting the performance of these putative diagnostic markers, given the heterogeneity of patient background in western and eastern cohorts (doi: 10.1002/hep4.1451; doi: 10.18632/oncotarget.24601).

Regarding HCC with or without cirrhosis, a study reported the dysregulation of three miRNAs in HCC biopsies depending on the presence or not of liver cirrhosis (DOI: 10.3892/ijo.2012.1716). Specifically, miR-24 and miR-27 resulted downregulated only in HCC patients with cirrhosis, whereas miR-21 resulted upregulated only in HCC patients without liver cirrhosis. This might explain the fact that the oncomiR-21, which is upregulated in most of cancer types including HCC, was not present neither in our miRNA-signature nor in the Murakami study where HCC patients with cirrhosis were considered. Similarly, we showed that three HCC-specific lncRNAs (CASC9, LUCAT1 and LINC01093) were deregulated in HCC tissues with respect to cirrhotic livers. We also showed a trend towards a decrease of CASC9 and LUCAT1 starting from healthy liver to cirrhosis without HCC, to cirrhosis complicated by HCC, in line with its possible contribution to hepatocarcinogenesis. In line, microRNAs deregulation may occur in surrounding cirrhotic livers as we showed for miR-30 family members being decreased starting from normal livers to cirrhotic livers with HCC to HCC (doi: 10.1158/0008-5472.CAN-19-0472).

The choice of the control group (healthy livers versus cirrhotic livers versus non-cirrhotic livers) is a relevant aspect to be considered when comparing results among different studies and is likely one of the reasons that render scientific reproducibility difficult in this disease. We showed that only a 6-miRNA signature is common between HCC arisen on cirrhotic and non-cirrhotic livers (miR-532, miR-34a, miR-93, miR-149#, miR-7f-2#, and miR-30a-5p). On the contrary, the expression levels of other miRNAs changed significantly between HCCs occurred on cirrhotic or non-cirrhotic backgrounds, making the control used for normalization a relevant factor and the standardization of controls crucial among studies (doi: 10.1007/s10620-017-4654-3).

We thank the Reviewer for this interesting comment. We have discussed these aspects in the text (pages 3, 5, 9, 22) and added four new References (Ref. 18, 38, 128, 129).